# Ohm's law lost and regained: observation and impact of transmission and velocity zeros

Krishna Joshi[1,2] ✉, Israel Kurtz[1,2], Zhou Shi[1,2,3] & Azriel Z. Genack ●[1,2] ✉

The quantum conductance and its classical wave analogue, the transmittance, are given by the sum of the eigenvalues of the transmission matrix. However, neither measurements nor theoretical analysis of the transmission eigenchannels have been carried out to explain the dips in conductance found in simulations as new channels are introduced. Here, we measure the microwave transmission matrices of random waveguides and find the spectra of all transmission eigenvalues, even at dips in the lowest transmission eigenchannel that are orders of magnitude below the noise in the transmission matrix. Transmission vanishes both at topological transmission zeros, where the energy density at the sample output vanishes, and at crossovers to new channels, where the longitudinal velocity vanishes. Zeros of transmission pull down all the transmission eigenvalues and thereby produce dips in the transmittance. These dips and the ability to probe the characteristics of even the lowest transmission eigenchannel are due to correlation among the eigenvalues. The precise tracking of dips in the conductance by peaks in the density of states points to a further correlation between zeros and poles of the transmission matrix. The conductance approaches Ohm's law as the sample width increases in accord with the correspondence principle.

Ohm's law, $V = IR$, and the geometric scaling of conductance, $G \equiv 1/R = A\sigma/L$, follow from the particle diffusion model[1]. Here, $A$ is the cross-sectional area, $\sigma$ is the conductivity, and $L$ is the sample length. The underlying quantum nature of conductance is revealed, however, in magnetoresistance fluctuations due to the interference of multiply-scattered, but still temporally coherent, electronic waves in micron-sized conductors at ultralow temperatures[2–10]. The scale of such mesoscopic conductors is intermediate between the microscopic atomic and the macroscopic realm with sample length longer than the transport mean free path and shorter than the inelastic scattering length, $\ell < L < L_{inel}$. For classical waves, such as light and sound, this condition is readily achieved even in macroscopic media, since the dimensions of the wavelength and scattering elements are much greater than the atomic scale so that quantum fluctuations do not affect the phasing of partial waves.

The variance of conductance in diffusive mesoscopic samples is nearly independent of sample dimensions[5–8]. Such universal conductance fluctuations (UCF) are a consequence of the correlation of flux across the sample brought about by the crossing of partial waves of electron quasi-particles[3,6,8–10]. In weak localization, constructive interference of partial waves following looped segments in opposite senses enhances the return to points within the medium and thereby suppresses the average conductance[3,4]. The suppression is greatest in the absence of an applied magnetic field when phase differences between waves following time-reversed paths in a loop due to the Aharanov-Bohm effect do not arise.

[1]Department of Physics, Queens College of the City University of New York, Flushing, New York 11367, USA. [2]Physics Program, The Graduate Center of the City University of New York, New York, New York 10016, USA. [3]OFS Labs, 19 School House Road, Somerset, NJ 08873, USA. ✉e-mail: krishnaaj078@gmail.com; agenack@qc.cuny.edu

Transport remains diffusive as long as the probability that partial waves following meandering Feynman paths of scattered electron quasi-particle waves loop back to a coherence length of the trajectory of one half the wavelength is low. Once the number of crossings of a typical coherence length by waves transiting the sample exceeds unity, waves are exponentially localized within the sample and conductance falls exponentially with sample length[3,11].

Landauer showed that electronic conductance is analogous to the classical wave transmittance, $T$, with $G = \frac{I}{V} = \frac{e^2}{\hbar} T$, where $\frac{e^2}{\hbar}$ is the quantum of conductance[12]. In the multichannel wire geometry, the transmittance is the sum of transmission coefficients between all the $N$ channels on the incident and outgoing surfaces of the conductor, $T = \sum_{a,b}^{N} |t_{ba}|^2$, where the $t_{ba}$ are the elements of the transmission matrix (TM)[13–24]. The transmittance can also be expressed as the sum of transmission eigenvalues $\tau_n$, which are the eigenvalues of the matrix $tt^\dagger$, $T = \mathrm{Tr}(tt^\dagger) = \sum_n^N \tau_n$[10,13,14,17]. The ensemble average of the transmittance is denoted as the dimensionless conductance, $g \equiv \langle T \rangle$. Dorokhov showed that each of the transmission eigenvalues scales with a different localization length[14]. In diffusive samples, in which $N/2 > g > 1$, there are $g$ "open" transmission eigenchannels (TEs) with $\tau_n > 1/e$ that carry most of the flux. The remaining TEs are "closed" with $\tau_n$ falling approximately exponentially with channel index, $n$[8,14–17,21,25–27].

The wave nature of propagation is immediately evident in the speckle pattern of light due to the interference of randomly phased partial waves passing through the medium[28]. The fluctuations of the speckle pattern with frequency shift reflect the length of optical paths through the medium, while their spectral correlation gives the time of flight distribution of pulsed transmission[29]. As the probability that a wave will loop back to cross a typical coherence length along its trajectory increases in the crossover to Anderson localization[3,8,9,30–32], fluctuations and correlation of transmitted flux are enhanced[32,33]. Anderson localization was predicted for electrons in a random lattice, but observed for microwave radiation[30], light[34] and sound[31].

The average transmission through a wedged sample falls inversely with sample thickness[35] in accord with particle diffusion theory[1] that yields Ohm's law. However, the wave nature of the conductance is observed in laterally confined regions without scattering. Quantum transport is manifest in the stepwise increase in the conductance of ballistic heterojunctions with gate voltage that controls the width of a region of free propagation[36,37]. The conductance jumps to a new plateau each time a new propagating wave channel is allowed. Analogous steps in transmittance are observed in the flux of diffuse light as the width of an aperture increases[38]. Similarly, weak localization is also observed for light in an enhanced retroreflection peak due to coherent backscattering[39–41], which reduces transmission.

The impact of disorder on the scaling of conductance has been explored in theory and simulations in diffusive samples[42–46] and in measurements on graphene nanoribbons[47]. A series of dips in the conductance is found as channels are closed as the Fermi energy is varied for different values of disorder and interpreted in terms of disorder-induced hybridization of subbands[48,49]. The dips found in coherent-potential approximation (CPA) calculations in narrow samples are smoothed as the disorder or sample size increases[48].

Despite the intensive study of mesoscopic fluctuations in the conductance of single samples, measurements of the ensemble average of conductance in diffusive mesoscopic samples have not been reported. Here we measure the microwave TM and find dips in transmittance near channel crossovers due to the correlation of vanishing transmission in the lowest transmission eigenchannel with dips in transmission of the high-transmission eigenchannels. Zeros of transmission arise when either the energy density vanishes at the sample output at topological transmission zeros (TZs) or when the eigenchannel velocity vanishes at a channel crossover. Dips in the conductance are also correlated with peaks in the density of states (DOS)

as a result of the correlation of zeros and poles of the TM. The correlation within the TM reduces the noise in the determination of the properties of TEs and thereby makes it possible to study the approach to vanishing transmission in spectra of the lowest TEs of 64-channel waveguides at levels many orders of magnitude below the noise in the measurement of the TM.

In this work, we reveal new aspects of global correlation within the TM and show that a full description of the scaling of conductance requires a parameter that captures the proximity to a channel crossing in addition to the conductance itself. A more complete understanding of global correlation would allow fuller retrieval of information carried by multiply scattering waves. This might find application to imaging and to data compression via dimensionality reduction[23]. The observation of the approach to zeros of transmission opens up possibilities for ultrasensitive detection of change within a medium. Two single zeros may interconvert to a complex conjugate pair of zeros[50,51] in a square root singularity as the sample is deformed[50,51]. Transmission also vanishes at a channel crossing because the longitudinal velocity of the newly introduced eigenchannel vanishes. This is reminiscent of the Wigner cusp in which the nuclear scattering cross section increases sharply near energies at which new scattering channels enter[52] and of optical analogues with potential for application to optomechanical accelerometry[53].

## Experimental results

The experimental setup used to measure spectra of the TM of random dielectric waveguides is shown schematically in Fig. 1a. The sample and experimental method are described in Methods. The square amplitude of the transmitted field for a particular configuration, frequency, and orientation of the source and receiver antennas on the measurement grid on the output surface is shown in Fig. 1a. The continuous speckle pattern of flux on the sample output determined for the same incident field as in Fig. 1a is shown in Fig. 1b. It is obtained from the amplitude squared of the fit of a sum of waveguide modes to the field pattern underlying the energy density in Fig. 1a. This enables a transformation of the matrix of field transmission coefficients between points on the input and output to the TM with a basis of the propagating modes of the empty waveguide.

Averaging over the transmittance measured for random configurations in three ensembles with identical composition but different lengths give the modulated spectra of the conductance, $g$, shown in Fig. 1c. Since $N/2 > g > 1$, the wave is diffusive for all sample lengths shown. The vertical dashed line at 14.8317 GHz represents the crossover from $N = 61, 62$ to $N = 63, 64$ channels. The new channels that enter are the degenerate transverse electric and transverse magnetic modes of the empty waveguide (Supplementary Note 1). Spectra of the transmittance for three random sample realizations with $L = 23$ cm and the variance of the transmittance over the ensemble measured are shown Supplementary Fig. S1a, b. Simulations of diffusive waves in Supplementary Fig. S1c show $\mathrm{var}(T) \cdot 2/15$ in the diffusive range in accord with UCF[6,9].

The source of the dips in conductance is revealed in the spectra of the individual transmission eigenvalues and their factors for a sample of length $L = 23$ cm: the longitudinal energy density, which is the energy density per unit depth, at the sample output, $u_n(L)$, and the longitudinal eigenchannel velocities (EVs), $v_n(L)$[54],

$$\tau_n = u_n v_n \qquad (1)$$

The reference to the sample output at $L$ is suppressed in the variables on the right-hand side of Eq. (1) to simplify the notation. Here, $u_n$ is proportional to the sum of the squares of the magnitude of electric field of the transmission eigenchannels over the output face of the sample, while $v_n$ is the average of the group velocities of waveguide modes comprising the TE weighted by their square

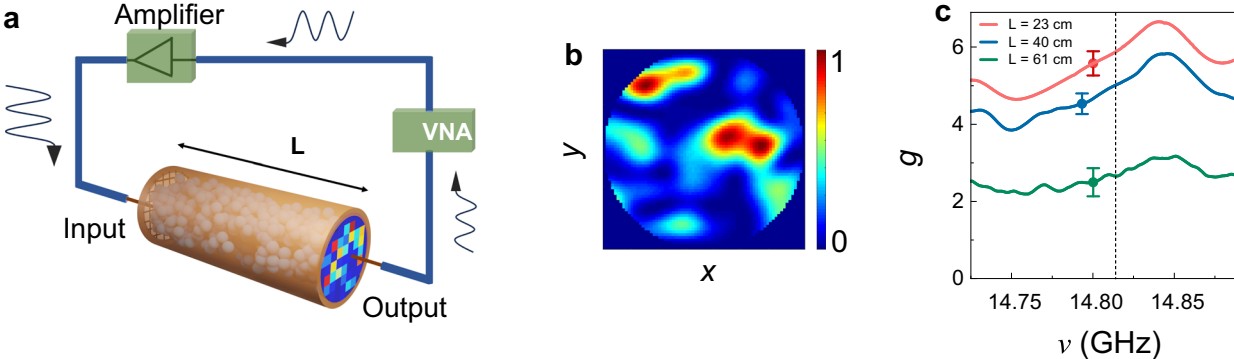

**Fig. 1 | Measurements of conductance. a** Microwave experiments are performed with use of a vector network analyser (VNA). The square of the field measured on all points on a grid on the sample output for a single position and polarization of the source antenna is shown. **b** The flux speckle pattern shown is obtained by taking the amplitude squared of the fit of a superposition of waveguide modes to the field on the grid of points shown in (**a**). **c** Measurements of spectra of the average transmittance, $\langle T \rangle \equiv g$, over 23 configurations for $L = 23$ cm, and 6 configurations for $L = 40$ and 61 cm. The error bars are square root of var $T = 2/15$ divided by the square root of the number of configurations.

amplitude. The procedures by which $u_n$ and $v_n$ are determined from measurements, as described above, are consistent with Eq. (1), as shown in Supplementary Note 2. The velocities of TEs are not randomized[54] and the ensemble averages of the EVs decrease as $\tau_n$ decreases, as seen in Fig. 2h.

Spectra of each of the variables in Eq. (1) are plotted for a single configuration of length 23 cm in Fig. 2a–d and for the average over 23 configurations in Fig. 2e–h. Most of the impact of absorption is compensated for numerically adding a gain of $\gamma_g = 0.01$ ns$^{-1}$ as described in Supplementary Note 3 and Fig. S11. Transmission spectra highlighting open and closed channels are displayed in linear and semi-logarithmic plots, respectively, in Fig. 2a, b. A series of precipitous dips in $\tau_{N=62}$ before the crossover from $N = 61, 62$ to $N = 63, 64$ is shown, together with a sharp rise from ultra-low transmission precisely at the crossover for $N = 63, 64$ (Fig. 2b). The smallest value of transmission in the spectrum of $\tau_N$ shown in Fig. 2b is more than a factor of $10^8$ below the average transmission in $\tau_1$ of nearly unity, while the signal-to-noise ratio (SNR) determined from the repeatability of measurements on the time scale needed to complete the measurement of the TM for a single configuration of 40 hours is 280 (Supplementary Note 4 and Fig. S4). Determination of the values of dips in transmission well below the experimental noise level reflects correlation among the transmission eigenvalues. The minimum value of $\tau_N$ found in the experiment is limited by the frequency step in the measurement of 300 kHz and by residual absorption. Values of $\tau_N$ as small as $10^{-12}$ are found in samples of length $L = 40, 61$ cm (Supplementary Fig. S2). The sharpness of the dips in spectra of the lowest transmission eigenvalue suggests that they are associated with TZs[50,51], which have not been previously been observed in experiments.

**Transmission zeros**

Transmission is predicted to vanish whenever zeros of the determinant of the TM appear on the real axis of the complex frequency plane. The determinant of the TM can be expressed as $\det(t) \sim \frac{\prod_{i=1}(\omega - \eta_i)}{\prod_{m=1}(\omega - \lambda_m)}$, where the $\eta_i = \omega_i \pm i\zeta_i$ are the zeros of $\det(t)$ and $\lambda_m = \omega_m - i\gamma_m$ are the poles[51]. The poles indicate the resonances or quasi-normal modes of the medium at frequency $\omega_m$ and half linewidth $\gamma_m$. Like the poles, the TZs are singularities in the map of the phase of the determinant of the TM, $\arg(\det(t))$, in the complex frequency plane. However, unlike the poles, which may appear anywhere in the lower half of the complex plane for unitary media, the TZs are constrained to appear only on the real axis or as complex conjugate pairs in the complex frequency plane[50]. In media in which the field decay rate due to absorption or gain, $\pm\gamma$, is

uniform, the entire phase map is shifted by $\pm\gamma$, so that TZs can be shifted to the real axis by adding absorption or gain.

The spectrum of $\log \tau_{N=62}$ before the crossover from $N = 61, 62$ to $N = 63, 64$ channels (Fig. 2b) is mimicked by the spectrum of $\log u_N$ (Fig. 2c). This is consistent with Eq. (1) since the spectrum of $v_{N=62}$, shown in Fig. 2d, does not exhibit singular behaviour before the crossover. A quantitative check of the agreement of measurements with Eq. (1) is made by comparing $\tau_{n=N=62}$ found from direct analysis of the measurement of the TM in Fig. 2b with the product of $u_{62}$ and $v_{62}$ in Fig. 2c, d. These are found, respectively, from the square magnitude of the transmitted field and from the weighted average of waveguide velocities for the 62$^{nd}$ TE. The plot of $\tau_N$ determined from the right-hand side of Eq. (1), shown as the gold circles in Fig. 2b, overlaps the curve determined from the direct analysis of the TM, confirming Eq. (1) and the accuracy of the determination of the parameters in the equation.

In addition to the sharp drops in $\tau_{62}$ before the crossover, the transmission eigenvalues of the new TEs that enter at the crossover vanish precisely at the crossover to $N = 63, 64$ (Fig. 2b, f). $\tau_{63}$ and $\tau_{64}$ rise sharply after the crossover as $v_{63}$ and $v_{64}$ increase from zero, while $u_{63}$ and $u_{64}$ enter with values above those for $u_{61}$ and $u_{62}$ before the crossover, as seen in Fig. 2c. Even though the contributions of closed channels to the transmittance are minimal, the appearance of zeros in transmission before and at a crossover is correlated with dips in the open channels, and so with dips in the conductance.

Further differences in the behaviour of zeros in transmission before and at the crossover emerge in a comparison of spectra of the properties of TEs in a single configuration (Fig. 2a–d) with the corresponding spectra of the ensemble averages (Fig. 2e–h). The sharp dips in $\tau_n$ before the crossover in single configurations disappear in the ensemble average, showing that the frequencies at which dips occur are random. However, the vanishing of $v_{63}$ and $v_{64}$ and the associated sharp rises in $\tau_{63}$ and $\tau_{64}$ precisely at the crossover survive averaging. We note that in some configurations sharp dips appear in $\tau_{64}$ after the crossover (Fig. S2). This indicates that the density of TZs is relatively high just before the crossover but does not entirely vanish after the crossover. The density of TZs becomes more uniform as the sample length increases, as seen in measurements (Fig. S2) and simulations (Fig. S13).

The dips in $\tau_N$ seen in Fig. 2b are examined more closely in the linear plots in Fig. 3. In addition to the spectra utilizing the added gain $\gamma_g = 0.01$ ns$^{-1}$ applied to the raw data of the TM in Fig. 2, spectra based on the raw data and with $\gamma_g = 0.0125$ ns$^{-1}$ are shown in Fig. 3a. The four numbered dips are due to TZs. The gain introduced in the

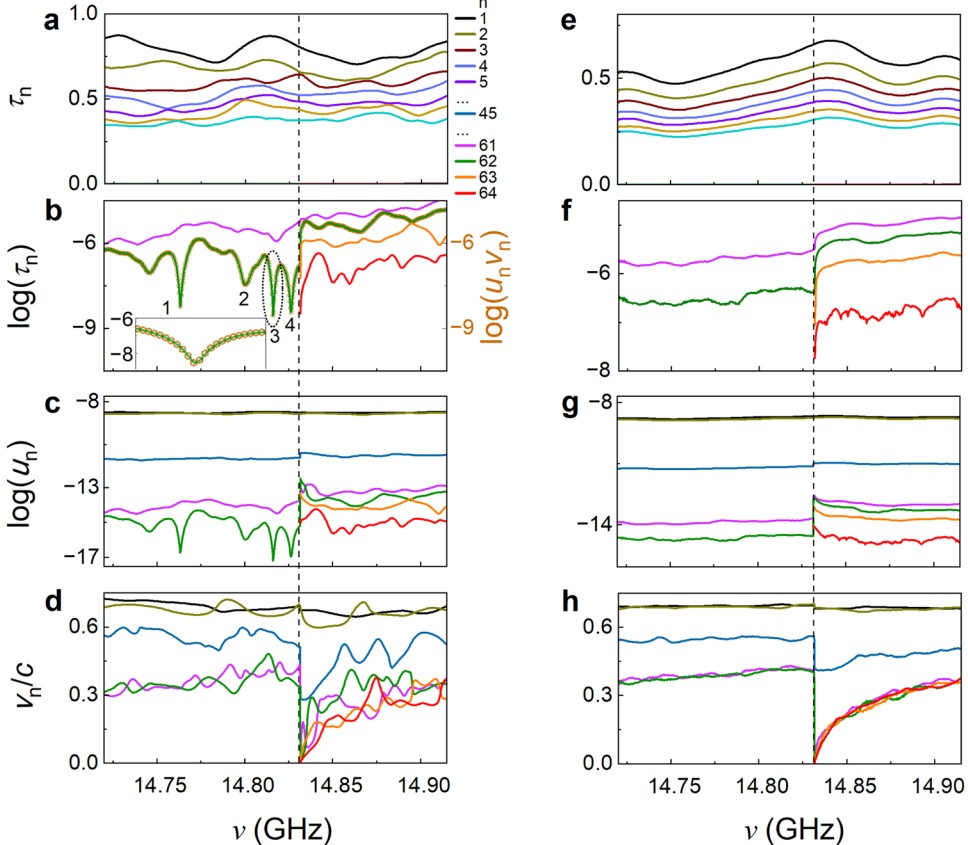

**Fig. 2 | Transmission eigenvalues and their factors obtained from an analysis of the TM.** The dashed vertical lines in (**a–h**) are at the crossover from $N = 61$, 62 to $N = 63$, 64 at $\nu_0 = 14.8317$ GHz. **a, b,** Linear (**a**) and semi-logarithmic (**b**) plots of spectra of $\tau_n$ for a single random configuration, showing, respectively, open and closed transmission eigenvalues. The sharp dips in the spectrum in green of $\tau_{N=62}$ before the crossover are associated with TZs. An analysis below and in

Supplementary Note 5 of the transmission time in Fig. 5f shows that dips 1, 3, and 4 are due to single zeros, while dip 2 is due to a complex conjugate pair of zeros. The gold circles are the logarithm of the product of measurements of $u_{62}$ and $v_{62}$, as shown in (**c**) and (**d**). **e–h** The configurational average values of $\tau_n$, log $\tau_n$, log $u_n$, and $v_n$. Only the dips in log $\tau_{63}$ and log $\tau_{64}$ and in $v_{63}$ and $v_{64}$ at the crossover survive averaging over configurations.

raw data counters the impact of absorption and thereby brings the single TZs, 1, 3, and 4, closer to the real axis in the complex frequency plane and lowers these minima in transmission. The dip at 2 is due to a conjugate pair of TZs. The nature of these dips is established in an analysis of the impact of gain upon the transmission time of the lowest TE shown in Fig. 5f. The quadratic fits of $A(\nu - \nu_0)^2$ to spectra of $\tau_{62}$ near the local minima at $\nu_0$ at the frequencies 1, 3, and 4 shown in Fig. 3b–d are in accord with simulations of spectra of $\tau_N$ around a TZ[51], and show that the spectra are not dominated by noise even near the minimum in spectra of $\tau_N$. Because absorption is not uniform within the sample, slightly different values of gain are applied near different minima in $\tau_N$ to achieve the lowest transmission at the minimum. The noise in measured spectra increases as additional gain is added, as demonstrated in Supplementary Note 3 and Fig. S11. This limits the range over which it is possible to compensate for absorption.

It is only possible to reliably determine spectra of $\tau_N$ from measurement of the TM because the corresponding noise level is many orders of magnitude below the level in the measurement of the TM. In Fig. 4a, b, we compare simulations of spectra of $\tau_N$, where $N = 15$, in a single configuration without and with added white noise in the $t_{ba}$ which gives SNRs of $10^5$ and $10^3$, respectively. The recursive Green's function simulations[55] used here are discussed in Methods and a model of the sample is shown in Supplementary Fig. S3. The noise in $\tau_N$ is proportional to the level of noise added in the elements of the TM but is much smaller. The standard deviations, $\sigma_n$, of the noise in $\tau_n$ for two

levels of added noise in Fig. 4a, b is seen in Fig. 4c to fall with increasing index of the TE. The plots of the SNR are precisely proportional and differ by the ratio of the SNRs of 100. The variation of $\sigma_n$ with TE index $n$ is seen to be well fit by a segment of a Gaussian function, $\sigma_n = \frac{A}{\sqrt{2\pi\sigma^2}} e^{\left(-\frac{(n-\mu)^2}{2\sigma^2}\right)}$, with $\sigma = 6.8$ and $\mu = -0.77$. The value of $A$ is proportional to the added noise. The inverse of the SNR of spectra of $\tau_n$ for both levels of added noise is seen in Fig. 4d to increase with TE index $n$. However, even for $n = N = 15$ and an SNR in $t_{ba}$ of $10^3$, the inverse of the SNR in spectra of $\tau_N$ is $5.8 \times 10^{-4}$. This low noise level is a consequence of the mutual repulsion of transmission eigenvalues which limits the range over which a given eigenchannel may vary[14,21,25–27].

The SNR in the measurement of the TM is 280 (Supplementary Note 4 and Fig. S4). This is due to the variation in temperature during the 40 hours over which the TM is measured. A measurement of the impact of temperature variation on spectra of transmission through a random mixture of polystyrene spheres is shown in Fig. S5. However, the hierarchy of TEs and their mutual repulsion is not disrupted by noise in the measurement of the TM. As a result, the noise level in $\tau_n$ is not set by the noise level in the measurement of the TM, and TZs can be identified in spectra of $\tau_N$. Further support for the identification of TZs is the similarity of spectra of $\tau_N$ and of the associated eigenchannel transmission time, $t_N$ derived from measurements and from simulations.

The identification of the dips in $\tau_N$ and $u_N$ (Fig. 2b, c) found from measurements of the TM with TZs is further supported by the

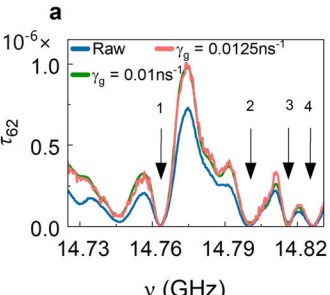 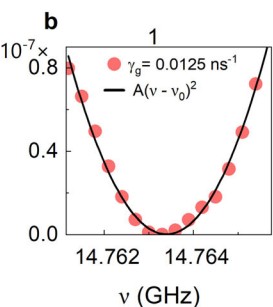 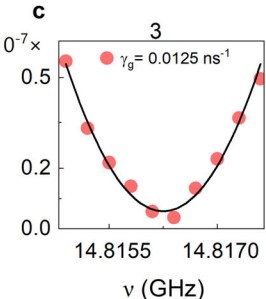 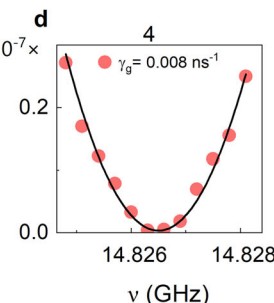

**Fig. 3 | Quadratic fits to minima in the spectrum of the lowest transmission eigenvalue. a** Spectra of the lowest transmission eigenvalue, $\tau_{62}$, before a crossover for the same configuration as in Fig. 2 determined from analysis of the TM for the raw experimental data (blue curve) and for spectra obtained by adding two different gain levels to the raw data are plotted in linear scale. Gain in the field at a level of $\gamma_g$ is added by Fourier transforming the raw spectra into the time domain and multiplying the field by $e^{\gamma_g t}$ before transforming back into the frequency domain[30]. The spectra exhibit four numbered dips due to transmission zeros (TZs). **b–d** The quadratic fits, $A(\nu - \nu_0)^2$, to spectra of $\tau_{62}$ near dips 1, 3, and 4 indicate that these are TZs[50,51]. The dip at 2 is due to a conjugate pair of TZs, as demonstrated below from the contribution of this feature to the transmission time for this TE.

similarity of these spectra to simulated spectra (Fig. 5a, b) and by the correspondence of those dips to singularities in simulations of the map of arg(det($t$)) in the complex frequency plane (Fig. 5d). Each of the dips in $\tau_7$ and $u_7$ in Fig. 5a, b, respectively, corresponds to a zero on the real axis of the phase map of det($t$) in the complex frequency plane shown in Fig. 5d. The zeros on the real axis are at the centres of the white circles and only occur before the crossover. A few of the poles are indicated with pink squares. The poles lie in the lower half of the complex plane and occur over the entire frequency range. EVs in the simulations shown in Fig. 5c behave similarly to the results of experiments in Fig. 2d; EVs only vanish at the frequency at which a new channel emerges. The vanishing of the EV at the crossover to a new channel produces a null in transmission which is not associated with a singularity in the complex plane. The crossover to the 8th channel in the simulations is set as the frequency of the 8th step in the transmittance of the empty waveguide. The crossover is not a multiple of $\lambda/2$ because of the discretization of the sample in the simulations. As the resolution in the simulation increases, however, the $N^{th}$ crossover approaches $N/(\lambda/2)$ (Methods and Supplementary Fig. S7).

In addition to the singularity in $\tau_N$ at a TZ, the transmission time $\tau_T$ in a unitary medium is singular when infinitesimal absorption or gain is added. This can be used to confirm that dips in $\tau_N$ correspond to TZs. $\tau_T$ is the sum of the transmission time of all the TEs, $\tau_T = \sum_n^N t_n$[51]. $\tau_T$ is the sum of Lorentzian functions associated with poles, $\tau_p$, and zeros, $\tau_z$, at angular frequencies $\omega_m - i\gamma_m$ and $Z_i + i\zeta_i$, respectively,

$$\tau_T = \frac{d(\arg(\det(t)))}{d\omega} = \tau_p + \tau_z = \sum_m \frac{\gamma_m}{(\omega - \omega_m)^2 + \gamma_m^2} + \sum_i \frac{\zeta_i}{(\omega - Z_i)^2 + \zeta_i^2}$$

(2)

In the absence of dissipation, the transmission time is proportional to the DOS and is the sum over Lorentzian lines associated only with the poles, $\tau_T = \tau_p = \sum_m \frac{\gamma_m}{(\omega - \omega_m)^2 + \gamma_m^2} = \pi\rho$[51,56]. This requires that the contribution of the zeros to the transmission time vanishes in unitary media. Equation (2) then requires that the zeros either lie on the real axis, $\zeta_i = 0$, or be complex conjugate pairs. This creates dips (peaks) in $\tau_T$ even at the smallest level of absorption (gain) as TZs on the real axis move into the lower (upper) half of the complex frequency plane.

The contributions of TZs to $\tau_T$ at low levels of absorption are predominantly represented in $t_N$[57]. The poles that contribute to $t_N$ are far off resonance[57] and thus contribute a nearly flat background to the spectrum of $t_N$. Consequently, the spectral features that appear in $t_N$ when low levels of gain or loss are introduced are due to zeros of det($t$) near the real axis in the complex frequency plane. Dips in $\tau_N$ in unitary

media, shown in Fig. 5a, are associated with dips in $t_N$ in Fig. 5e when small absorption is added. Each of the dips in $t_N$ are nearly Lorentzian functions with depth $1/\gamma_i$ and half-width $\gamma_i$, in accord with Eq. (2), as shown in Supplementary Note 5 and Fig. S12. The different depths of the dips are due to the different average field absorption rates in different quasi-normal modes in media with nonuniform absorption. Sharp features in the experimental spectrum of $t_{62}$ in Fig. 5f appear at the same frequencies as the dips in transmission for $\tau_{62}$ for the configuration shown in Fig. 2b. Adding gain numerically to the field spectra from which the TM is constructed modifies the spectrum of $t_{62}$ in a manner consistent with a TZ near the real axis moving vertically in the complex plane.

### Global correlation

The association of dips in the conductance near channel crossovers with reduced transmission in the lowest TE is buttressed in a comparison of spectra of $g$ (top panel of Fig. 6a) with spectra of the probability density of TZs, $\rho_0$, and of the average velocity of the lowest TE, $v_N$, in simulations of the crossovers to the 10th and 11th channels (middle panel of Fig. 6a). $\rho_0$ is peaked right before a crossover, while $v_N$ vanishes at the crossover and rises slowly afterwards. The relatively sharp drop in $g$ before the crossover and its slower recovery after the crossover bears similarity to the rise in $\rho_0$ before the crossover and the slower recovery of $v_N$ from zero after the crossover (second panel in Fig. 6a).

Beyond the qualitative associations of the dips in $g$ with characteristic features of the lowest TE, there is a striking quantitative relation between dips in $g$ (top panel of Fig. 6a) and peaks in the DOS, $\rho$ (bottom panel in Fig. 6a). The dips in $g$ are replicas of the peaks in $\rho$ (top panel of Fig. 6a). Similar behaviour is seen in microwave measurements shown in Supplementary Fig. S8. Expressing $g$ and $\rho$ as sums of a slowly varying term and a remainder term which is modulated on the scale of the change in width or frequency at which new channels are introduced, $g = \bar{g} + \Delta g$, and $\rho = \bar{\rho} + \Delta\rho$, respectively, the remainder terms are proportional

$$\Delta g = -\alpha\Delta\rho \tag{3a}$$

$$g = \bar{g} - \alpha\Delta\rho \tag{3b}$$

The slope of $\bar{g}$ is the average of the slopes of lines passing through the local minima and maxima of the respective functions. An excellent fit of the right-hand side of Eq. 3b to $g$ in terms of the parameter $\alpha$ is shown in the upper panel of Fig. 6a.

The impact of TZs and low values of $v_N$ upon the conductance in samples of the same length but more than six times the width as in

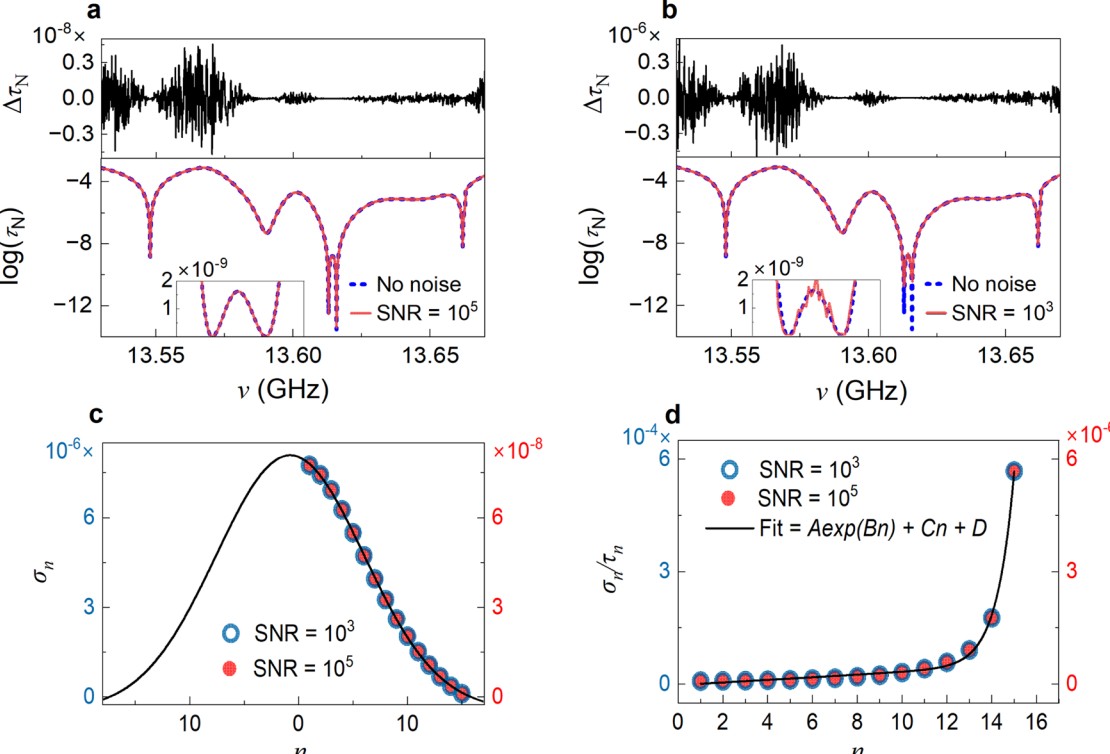

**Fig. 4 | Impact of white noise upon spectra of transmission eigenvalues.**
**a** Simulations of the spectrum of $\tau_N$ with and without added white noise (lower panel) with SNR of $10^5$ in the real and imaginary parts of $t_{ba}$ (SNR of $5 \times 10^4$ for the flux $|t_{ba}|^2$) for a single random configuration with $N = 15$, and a plot of the deviation of this spectrum from the spectrum of $\tau_N$ without added noise (upper panel). Dips at TZs of $10^{-13}$ are observed. **b** The spectrum of $\tau_N$ in the same configuration as in (**a**) but with a 100-fold increase in the noise level. The typical change in the spectrum of $\tau_N$ is larger by a factor of 100 compared to the results in (**a**). The spectrum at the TZ falls only to $10^{-10}$. **c** The standard deviation, $\sigma_n$, of the noise in the $\tau_n$ vs. $n$ falls with increasing $n$. When multiplied by a factor of 100, the plot of $\sigma_n$ for SNR of $10^3$ overlaps that for a SNR of $10^5$. Excellent fits of $\sigma_n$ are shown for Gaussian functions for three different widths in Supplementary Fig. S6. **d** The inverse of the SNR in spectra of $\tau_n$, $\sigma_n/\tau_n$, increases with $n$.

Fig. 6a are seen in Fig. 6b to be greatly attenuated. This reflects the weakened correlation between the open and closed transmission eigenvalues (Fig. 6c and Supplementary Fig. S9). In order to present a broad range of transmission eigenvalues, the eigenchannels $\tau_n$ are parametrized in terms of variables $x_n$, with $\tau_n \equiv \cosh^{-2} x_n$, which are approximately equally spaced for $n < N/2$ [14,17,26]. In a sample with $N \sim 8$, correlation between $x_N$ and $x_1$ is substantial, whereas it is small for $N \sim 40$. The modulation of $g$ is also reduced because TZs no longer arise exclusively just before the crossover but occur over the entire frequency range as the residual peak in $\rho_0$ broadens. As a result of the decrease in correlation among transmission eigenvalues and the widening of the range of TZs, the modulation of $g$ is washed out in the limit of large $N$. The conductance then approaches a linear, Ohmic variation with sample width (Fig. 6d).

For a given sample length and scattering strength, the proportionality constant, $-\alpha$, between $\Delta g$ and $\Delta \rho$ is independent of sample width even as transport crosses over from being diffusive to being localized with $g$ falling below unity. $\alpha$ falls, however, as the sample length increases. The modulation found in simulations is shown in Fig. 7. Since the dip in $g$ is correlated with a high density of TZs, this suggests that poles and TZs are correlated. Perhaps this is because a high density of poles presents greater opportunities for destructive interference between modal field that produces a TZ.

## Discussion
Instead of the linear scaling of the conductance with sample width predicted by Ohm's law in multiply scattering media, and the stepwise increase with width found in ballistic samples [36–38], the transmittance measured in diffusive random waveguides is punctuated by dips as the frequency increases [42–49]. Measurements of the microwave conductance show that these dips are caused by the correlation between zeros in transmission in the lowest TE and transmission in open channels. When TZs are spectrally isolated, dips in transmittance coincide with the TZs (Supplementary Fig. S10). Transmission may vanish in the lowest TE both at TZs and at zeros of the EV at the crossover to a new channel. Spectra of the lowest TE in a waveguide supporting up to 64 channels are found even though the noise in the measurement of the TM greatly exceeds the signal in the lowest TE, and particularly near zeros of transmission. This is possible because the range of values of a given transmission eigenvalue is limited by their mutual repulsion and the correlation between them [17,24,26,27]. Further, the TM encapsulates the positions of zeros and poles in the complex frequency plane, and, therefore, the zeros in the spectrum of $\tau_N$.

Since the field in the medium, including at frequencies at which it may vanish, is determined by the constellation of poles in the phase map of the determinant of the TM in the complex plane, the poles and zeros are correlated. This correlation is seen in the proportionality of the spectrum of the dip in conductance to the rise of the DOS relative to the secularly increasing backgrounds. The origin of this proportionality and of the independence of the proportionality constant on sample width at a given length will be explored in future work. The opposite sign of the modulations of $g$ and $\rho$ are an invitation to investigate the Thouless relation, $g \sim \rho \delta \omega$, where $\delta \omega$ is the average linewidth of quasi-normal modes, generally assumed to be independent of sample width. Thus, $g$ and $\rho$ might be expected to be

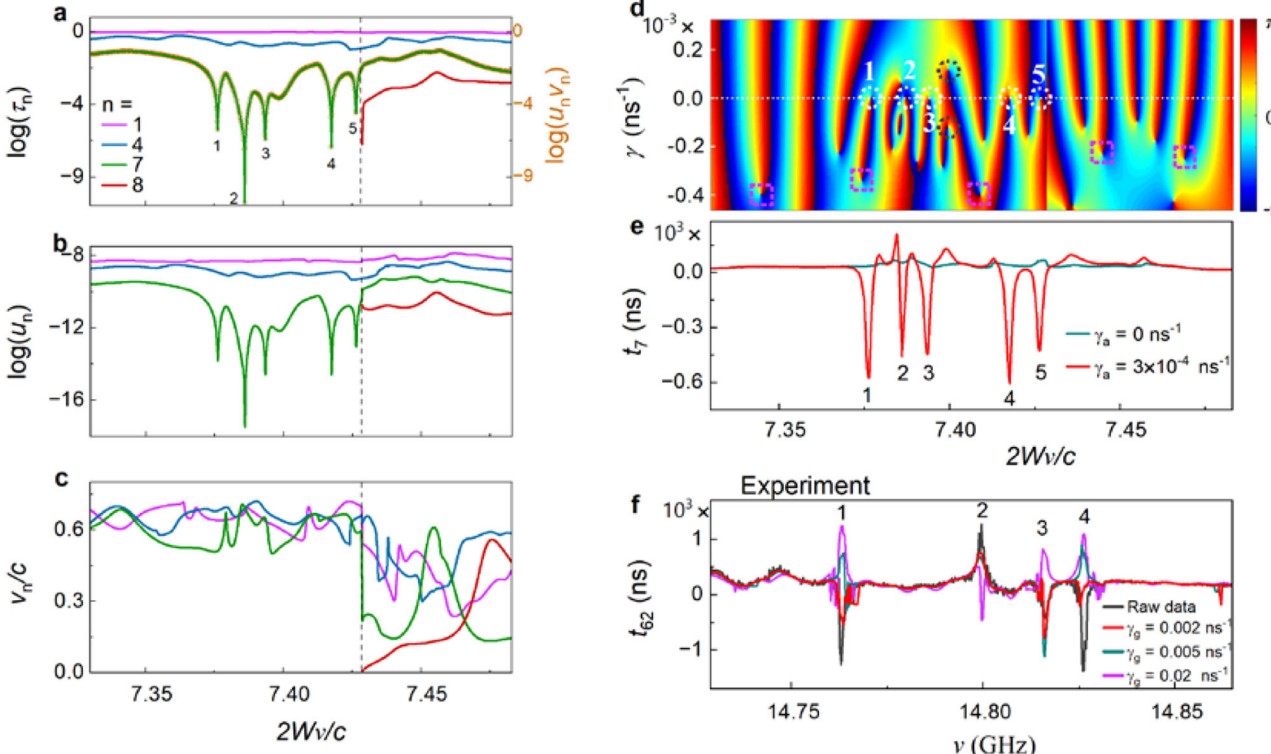

**Fig. 5 | Comparison of simulations of transmission eigenvalues and transmission zeros with measurements of transmission time.** Simulations of propagation in a medium of width $26a$ and length $1000a$ near the crossover from $N = 7$ to $N = 8$ channels. Here, $a$ is the length of the sides of the square elements of the sample. **a** Five sharp dips appear in spectra of $\log \tau_7$ before the crossover to the 8th propagating channel. The gold circles are the logarithm of the product $u_7 v_7$, with the factors shown in (**b**, **c**). **b** Spectrum of the logarithm of the linear energy density at the sample output, $u_n$, and of the $\tau_n$ are similar except at the crossover. **c** The EV of the new channel vanishes at the crossover and pulls down $\tau_8$. **d** Map of $\arg(\det(t))$ in the complex frequency plane. The horizontal axis is plotted in the dimensionless units $2W\nu/c = 2W/\lambda$, while the vertical axis represents the amplification rate of the field. Circles of white dots surround the TZs responsible for the five dips labelled in (**a**), while black dots circle the TZs of a conjugate pair of zeros. A few of the poles are indicated with pink squares. **e** Transmission time $t_n$ in a medium without absorption, $\gamma_a = 0$ (blue curve), and with absorption $\gamma_a = 3 \times 10^{-4}$ ns$^{-1}$ (red curve). **f** The eigenchannel transmission time for $N = 62$ based on the raw experimental data for the configuration shown in Fig. 2a–d and for three levels of added gain $\gamma_g$. The variation of $t_{62}$ with loss and gain near the frequencies of the dips in transmission in Fig. 2b is consistent with the impact of TZs on the transmission time, as given in Eq. (2).

proportional, but we find they change in opposite directions relative to a secularly increasing background. It may be that the Thouless relation holds for the secular part of $g$ and $\rho$, and is more properly expressed as $\bar{g} \sim \bar{\rho}\overline{\delta\omega}$, where $\overline{\delta\omega}$ is the average of the modal linewidth between crossovers. The modulation at channel crossovers may then reflects a quantum size effect, which is increasingly attenuated as the dimensions of the sample increase.

The correlation between low- and high-transmission eigenvalues falls as $N$ increases because the probability density of TZs broadens spectrally and because the correlation of transmission eigenvalues is of finite range. As a result, the transverse scaling of the transmittance approaches the linear scaling of Ohm's law predicted by the particle diffusion model. Thus, the classical result is approached as the quantum number $N$ increases.

In the absence of dissipation, a conjugate pair of TZs may approach and merge on the real axis at a zero point (ZP) when the sample is modified to produce two single TZs that subsequently move on the real axis[50]. The distance between singularities in the complex frequency plane has a square-root singularity versus modifications of the sample as a ZP is approached[50]. Observations of TZs as a sample are perturbed may, therefore, provide a means for ultrasensitive detection of changes in a sample.

The vanishing of the longitudinal component of velocity of the lowest TE, $v_N$, at a crossover is key to understanding the transition to a new channel. A quantitative study of the vanishing of the EV and a peak in the DOS at channel crossovers in a random

medium may shed light on singularities at the opening of new channels in nuclear[52] and optical scattering[53], and at grating anomalies[58].

## Methods

### Experimental setup, sample, and measurements

The experimental setup used to measure the spectra of the TM of random dielectric waveguides is shown in Fig. 1a. The samples are collections of randomly positioned alumina spheres of diameter 0.95 cm and refractive index 3.14 embedded in Styrofoam shells to produce an alumina volume fraction of 0.07 within a cylindrical copper tube of diameter 7.3 cm. Spectra of the in- and out-of-phase components of the transmitted phase relative to the incident field are obtained for random samples of lengths, $L = 23$, 40, and 61 cm with use of a vector network analyzer (VNA) over the frequency range of 14.70–14.94 GHz. The source and detector are separated from the sample surface by a 2-cm-long segment of empty waveguide. The wave is thus launched and collected from a length of empty waveguide and are not in close proximity to the sample.

The output of the VNA is amplified to a power of nearly 1 W. Field spectra are measured for each of two perpendicular orientations of the source and receiver antennas for each pair of locations on a 9-mm grid on the input and output surfaces of the sample. The antennas are the central conductors of copper clad cables, which extend beyond the cladding and are bent by 90° and aligned parallel to the sample's surface. The fields emitted and received at the input and output of the

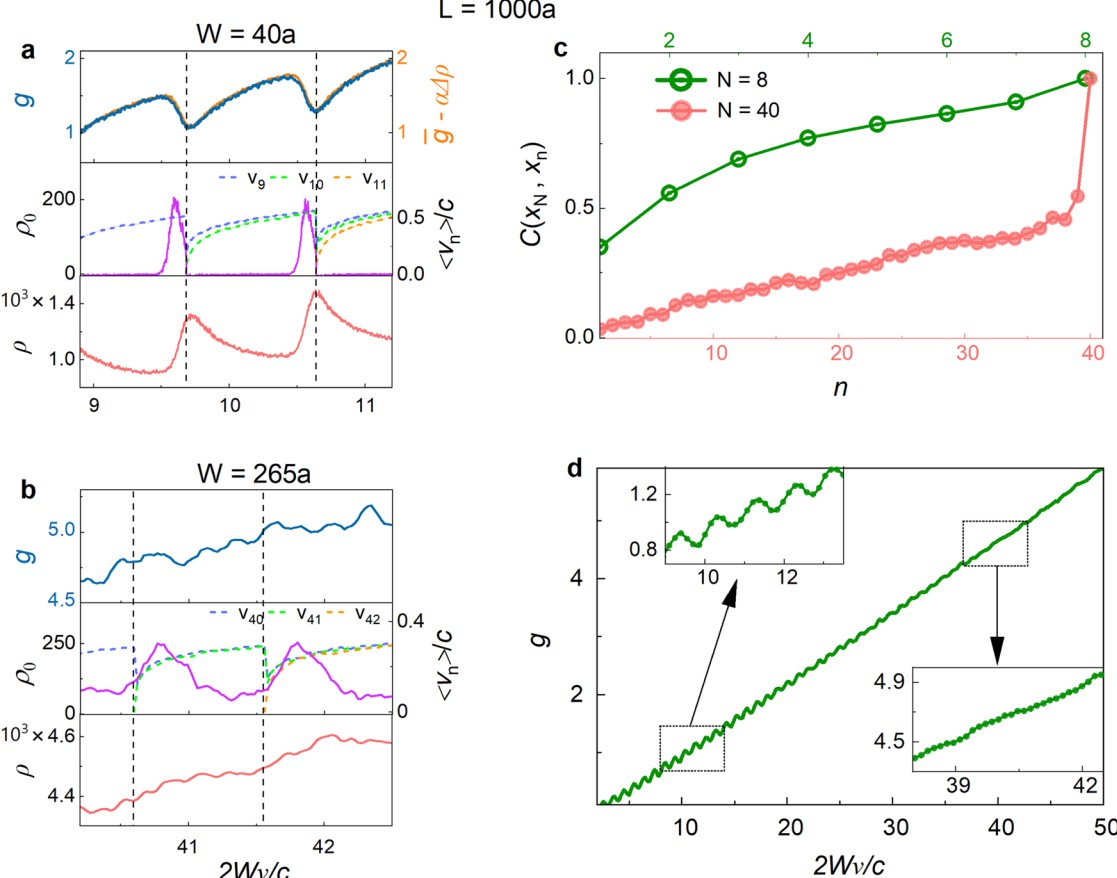

**Fig. 6 | Impact of transmission zeros, eigenchannel velocity zeros and the DOS on the dimensionless conductance. a** $W = 40a$; **b,** $W = 265a$, with $L = 1000a$. Simulations over 300 configurations of spectra of $g$, the density of TZs, $\rho_0$, the EVs of the lowest transmission eigenchannel, $v_N$, and the DOS, $\rho$, vs. $2W\nu/c$ in the vicinity of two channel crossovers for two ranges of sample width. The DOS is found from $\rho = \tau_T/\pi$. Spectral regions with a high density of TZs or low values of $v_N$ are seen to correspond to dips in $g$ and peaks in $\rho$. In the upper panel of (**a**) $g$ is fit by the sum of a slowly varying term $\bar{g}$ and a term proportional to the modulated part of $\rho$, $-\alpha\triangle\rho$, with $\alpha$ serving as the fitting parameter, in accord with Eq. (3b). The modulation of the DOS is diminished in wider samples, as seen in (**b**). The peaks in $\rho_0$ shift towards lower frequencies with respect to the crossover as the sample width

increases. Such a shift is seen in the measured conductance in Fig. 1c. **c** Degree of correlation between the lowest transmission eigenvalues and other channels. Eigenvalues $\tau_n$ are parametrized in terms of variables $x_n$, $\tau_n \equiv \cosh^{-2} x_n$ [14]. The spacing between the $x_n$ are approximately constant for the first $N/2$ transmission eigenvalues [26]. For the narrower sample with $N = 8$, the correlation of transmission in open channels with $x_N$ is high, while, for $N = 40$, the correlation is low. As a result, the depth of modulation of $g$ falls with sample width and excitation frequency as $N$ increases. **d** $g$ as a function of sample width at a frequency of 14.7 GHz. The rectangular insets show zoomed-in views of the curve with strong and weak modulation of $g$ for narrow and wide samples, respectively.

sample are polarized along the direction of the antennas. Because the TM is not complete, perfect transmission is not observed, but ultra-low transmission is still measured. New sample configurations are produced by rotating the sample about its axis and then vibrating the sample momentarily so that the sample settles. Field spectra are reproducible over the 40 h required for the measurements of the TM of a single realization of the disorder as discussed in Supplementary Note 4 and Fig. S4.

**Recursive Green's function simulations**
Simulations of electromagnetic wave propagation are carried out on rectangular samples comprised of a lattice of square boxes with random refractive index. The sides of the boxes have length $a = \frac{\lambda}{2\pi}$ and the refractive index $n = n_r + jn_i$, where the real part $n_r$ is drawn randomly from a rectangular distribution $[1 - \Delta n, 1 + \Delta n]$ with $\Delta n = 0.3$. The imaginary part $n_i$ is calculated from the decay rate $\gamma$ using the relation $n_i = \gamma n_r \lambda / 4\pi$. The values of $n_i$ differ among the cells so that the temporal decay rate $\gamma$ is constant. Supplementary Fig. S3 represents our computational model. The sample is connected to semi-infinite leads on the left and right and bounded on top and bottom by perfect reflectors. This is equivalent to the sample being uniform and

unbounded in the direction perpendicular to the plane of the rectangle. The field excited from the open boundaries of the sample may be expressed in terms of the waveguide modes of the empty waveguide with $n = 1$.

We employ the recursive Green's function method [54] to find the field in the $k^{th}$ column for a source on the left-hand side of the sample. The sample is divided into $K$ columns of width $a$, labeled $1, 2, \ldots, k, \ldots K$. We begin with the surface Green's function of the left lead and proceed from left to right, connecting each column to the subsystem, with columns 1 to $k - 1$ to its left. The coupling between a $k^{th}$ column and the subsystem to the left is expressed via the Dyson equation, $G = G_0 + G_0 V G$. $G$ is the Green's functions of the connected columns from 1 to $k$. $G_0$ is the matrix of Green's functions of the connected columns from 1 to $k - 1$ and the disconnected column $k$, and $V$ is the interaction terms between $k - 1$ and $k$. We ultimately find the result $G_{k1} = [1 - G_{kk}^L V_{k, k+1} G_{k+1, k+1}^R V_{k+1, k}]^{-1} G_{k1}^L$, which allows us to calculate Green's functions for all columns from 1 to $K$.

The unitarity of the scattering matrix (S) in a sample found $\|S^\dagger S - I\| \sim 10^{-8}$. The spatial distribution of the output fields, as determined by Green's function, is fitted with a superposition of waveguide modes, and the TM is expressed in waveguide mode space. TEs are

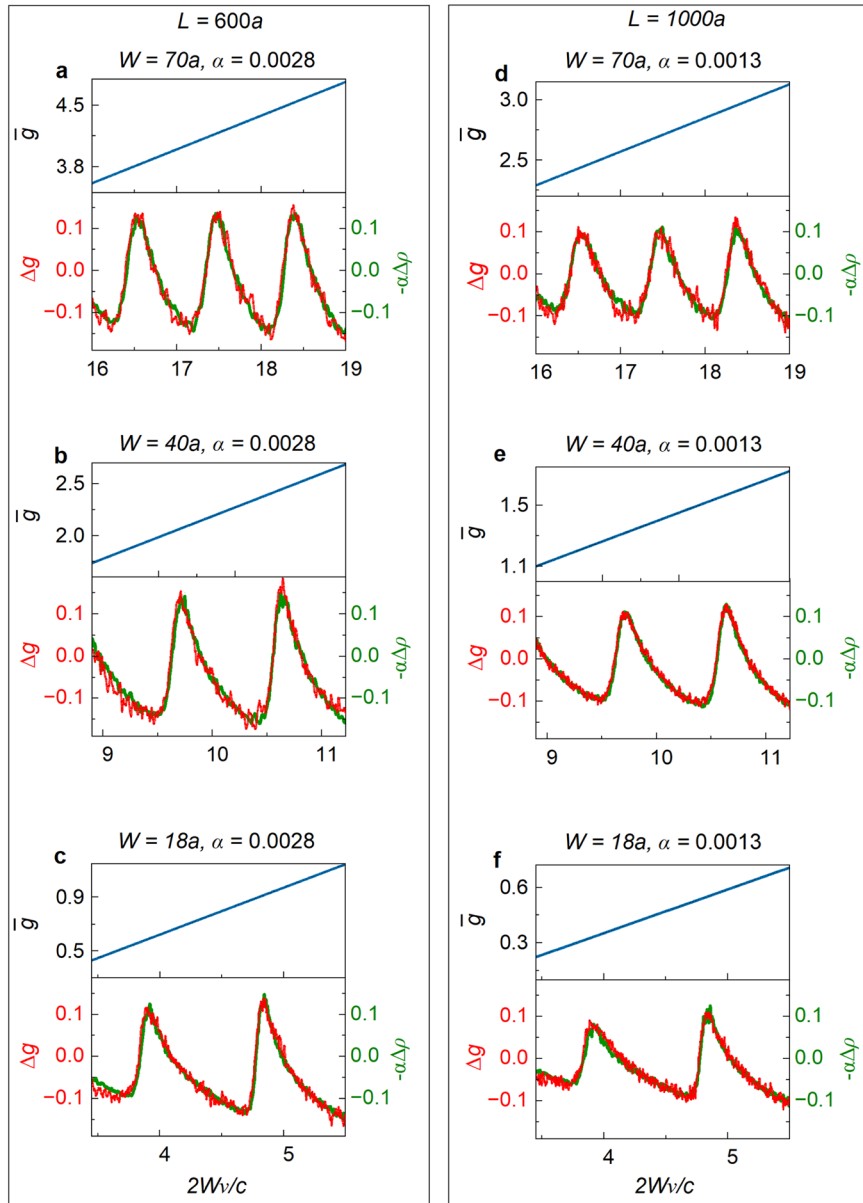

**Fig. 7 | Modulation of conductance and the density of states. a–f** Fits of $\Delta g$ by $-\alpha \triangle \rho$ through channel crossovers from simulations for 300 configurations for each of two sample lengths. The slowly varying contribution to the conductance, $\bar{g}$, is shown above plots of $\Delta g$ and $\Delta \rho$. $\alpha$ is seen to be independent of sample width for a given length and to fall with sample length.

determined using the singular value decomposition (SVD) of the TM, $t = \mathscr{U}\Lambda\mathscr{V}^{\dagger}$. Here, $\Lambda$ is a diagonal matrix whose elements are singular values, $\lambda_n = \sqrt{\tau_n}$. $\mathscr{V}$ and $\mathscr{U}$ are unitary matrices on the sample's input and output surfaces, respectively. The amplitude of the $m^{\text{th}}$ channel in the $n^{\text{th}}$ TE on the input and output surfaces are given by the $v_{nm}$ and $u_{nm}$. By projecting the TM onto the input vector $\mathscr{V}$, we obtain the energy density profile at the output surface: $u_n(L) = \sum_{m=1}^{N} |t_{nm} v_{mn}|^2 / v_{wm}$, where $v_{wm}$ is the group velocity of the $m^{\text{th}}$ waveguide mode.

**Impact of spatial resolution on onset of new channels in simulations**

Simulations of the wave equation for the electric field are carried out for a bounded mesh of cells. As a result, the crossover to a new channel does not occur at $N = W/(\lambda/2)$. The difference in the wavelength at the onset of a new propagating channel from $\lambda = W/(N/2)$ decreases as the size of the cells decreases and vanishes in the continuum limit. The calculation of the number of channels and the cause for the difference are outlined below.

For a discretized wave, $k_x^2 = 2(1 - \cos(\frac{m\pi}{W})) = 4\sin^2(\frac{m\pi}{W})$ and $\sin^2(\frac{k_z}{2}) = \frac{k^2 - k_x^2}{4} = \frac{k^2}{4} - \sin^2(\frac{m\pi}{2W})$, where $z$ is the direction of propagation and $x$ is the transverse direction. A sample of $m$ cells in the transverse direction generates a set of values $\frac{m\pi}{W}$ corresponding to the transverse $k$-vector of propagating modes that enter the system at a particular $\lambda$. But since $k_x < k$ propagating modes must satisfy the condition $\sin^2(\frac{k_z}{2}) = \frac{1}{4}(k^2 - k_x^2) > 0$. Higher simulation resolution corresponds to a greater number of cells of smaller size, generating a larger set of values $\frac{m\pi}{W}$ that give values of $N$ that are closer to being integers. Supplementary Fig. S7 shows the effects of resolution of the simulation on the accuracy of the value of $N$.

## Data availability

The datasets generated during and/or analysed during the current study are available at https://doi.org/10.6084/m9.figshare.25880413.

## Code availability

The simulation codes used in the current study are available from the corresponding author on reasonable request.

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

## Acknowledgements
The authors thank Yiming Huang and Asher Maor for valuable discussions. This work is supported by the National Science Foundation (US) under NSF-BSF Award No. 2211646 (AZG). Correspondence and requests for materials should be addressed to Azriel Z. Genack and requests for computational codes should be addressed to Krishna Joshi.

## Author contributions
A.Z.G. conceived and directed the project, Z.S. measured the microwave transmission matrix. K.J. and I.K. participated in the planning of the research, analysed the experimental data and carried out and analysed numerical simulations. A.Z.G. with input from J.K., I.K., and Z.S. wrote the manuscript.

## Competing interests
The authors declare no competing interests.
