## [Transparent Peer Review file · Nature Communications]

Ohm's law lost and regained: observation and impact of transmission and velocity zeros

Corresponding Author: Professor Azriel Genack

Version 0:

Reviewer comments:

Reviewer #1

(Remarks to the Author)

The authors of the current manuscript analyze experimentally and computationally the behavior of transmission eigenvalues in the proximity of a close-to-open channel threshold. They find that the minimal transmission eigenvalue vanishes at these frequencies, leading to dips in the transmittance of a diffusive sample. The experimental data are nicely compared with the numerical simulations and the main message is nicely conveyed. The results might be interesting for a large (and broad) scientific community – ranging from imaging to wireless communications.

The analysis relies on the connection between transmission eigenvalues with longitudinal energy density at the sample output and the longitudinal eigenchannel velocity. The equation that describes this connection is Eq. (1). Maybe I am missing something, but is this relation dimensionally correct?

Also, I feel that a bit more work is needed in the introduction. While the abstract and conclusions were pretty clear, the introduction is missing this clarity. Specifically, the last paragraph has to be re-written, the main result stated clearly and the potential connection to quantum electronics (and/or) other applications (e.g. imaging?) has to be highlighted.

I am ready to recommend publication of the manuscript to Nature Communications provided that the authors address these comments in their revision.

Reviewer #2

(Remarks to the Author)

In this article, the authors experimentally measured low transmission eigenchannels through a scattering waveguide and demonstrated that modulations in the conductance g are driven by zeros of the transmission matrix (TM) near the crossover to a new channel. This study shows that for a small total number of channels, the zeros of the TM break Ohm's law, which asserts that conductance scales linearly with width. Although the existence of conductance fluctuations has been known for decades, the major achievement of this paper is the detection of low transmission channels from correlations in the transmission matrix and the experimental demonstration of the connection between these channels and conductance fluctuations.

Overall, the results are technically sound and the link to conductance variations makes this work both compelling and impactful. The experiments were done thoroughly, the data were collected over a large number of disorder realizations, and of high quality as result of measurement stability and a solid methodology. Aside from a few minor comments, I recommend this manuscript for publication in Nature Communications.

Comments and Suggestions:

The authors should make it clear that transmission eigenchannels below the noise level are never experimentally measured, but rather they are detected indirectly via measurements of the transmission matrix. Because of this, lines 58-59 and 267-268 come off as a bit misleading.

It would be helpful to know what the noise level is for the experiment. For example, what is the smallest transmission

eigenchannel that can be experimentally measured if it is sent into the disordered waveguide?

For experimental results averaged over many disorder realizations (e.g., Fig. 1c) the authors should include the standard error in the plots. This will also provide some insight into the experimental noise.

Conceptualizing zeros and poles of the TM can be challenging, as low transmission eigenchannels are typically not realizable in experiments. Including a physical description for how these channels behave would be helpful, even if this behavior is not explicitly observable. The effect of poles on transport is particularly confusing. Can poles be brought to the real axis with sufficient gain and, if so, what would it physically mean for the transmission to diverge? Is this related to lasing as in the case of a pole for the scattering matrix?

The current title is somewhat confusing to general readers. Specifically, it is unclear what the zeros and poles refer to, making it hard to determine what this paper is about or worth reading. The authors are encouraged to provide a clearer title than the current one.

Hui Cao, Rohin McIntosh

Reviewer #3

(Remarks to the Author)

Reviewer #4

(Remarks to the Author)

The work by Joshi et al is a very interesting exploration of the occurrence, in microwave experiments as well as in numerical simulations, of near-zero eigenvalues of the transmission matrix in the diffusive regime.

Typically, the smallest eigenvalue of the TM is expected to be exponentially small and negligible with respect to the dominating eigenchannels which have transmittance of order unity. Here the authors show that near the onset of a new channel dips in the transmittance of this lowest channel occur to yield an even lower transmittance. The authors were not able to directly excite such channels to demonstrate the absence of transmitted power directly, the low transmittances were the result of an eigenvalue decomposition of the measured transmission matrix.

Here we find an unexplained and unelaborated weakness of the paper: The authors claim to resolve a channel with a transmittance that is nine orders of magnitude below the experimental signal to noise. They give no clue how this is possible and why we should not immediately discard the effect as an artefact of their data analysis procedure.

Moreover, their determination of the experimental SNR leads to many other questions. They claim an average value of 1300, but the 3 typical curves they show all have a SNR of 200 or less. And what is the average SNR anyway, $\langle N \rangle$, $\langle S/N \rangle$, or $1/\langle N/S \rangle$?

It seems that the procedure they use to determine the SNR mostly measures drifts due to e.g. mechanical or thermal creep in the sample, not the instantaneous SNR of the measurement device. So very likely this is a highly correlated form of noise, that could well cause artefacts in the data. A result of such artefacts could well be that transmission zeros that were originally not on the real axis are shifted onto it.

While the experimental work is very interesting, there are several statements in the manuscript that seem wrong, contradictory, circular, or imprecise, and that make me reluctant to advise in favor of publication. Therefore I recommend a thorough revision of the text.

Comments and questions on the procedure

First of all it should be explained how a signal that is nine (or six?) orders below the noise can be trusted. What checks did the authors do to make sure that transmission zeros really exist and are not artefacts of the noise/drifts that occur during their measurements?

What is the claim on the noise level anyway? L 267 speaks of 9 orders below the experimental noise level, while L 99 it is 9 orders between the signal level of the most open channel, and the SNR is claimed to be about 1000. So there is a factor 1000 inconsistency between these statements.

In the case of the simulations, the reference for the recursive Green's function method (42) does not seem to be the original one (45). This method, although powerful, can have numerical "gain" or "loss" due to rounding errors, did the authors check the unitarity of the S-matrix?

L102 "measurements of dips in transmission well below the experimental noise level reflect correlation between the eigenvalues" That sounds nonsensical as only one eigenvalue seems affected. They DO however reflect correlations

between the matrix elements that seem to (through some unexplained mechanism?) survive the noise.

The discussion on Ohms law (L255 and on) seems to apply just as well to the type of conductance modulation seen in fig 4a and already described long ago (e.g. DOI:<https://doi.org/10.1103/PhysRevB.40.5941>). Ohm's law has nothing to say about single transmission channels. Hence this discussion (and the first part of the title of the paper) seem beside the point. It is however interesting that the conductance drops occur around the same time as the transmission zeros.

L67 .. the continuous field can be recovered ... This suggests full recovery of the field is possible but L320 " the TM is not complete". How incomplete is the TM and how does this incompleteness relate to what is claimed on L67?

In the same line: The authors express the field in terms of the propagating modes of the empty waveguide. However if scatterers are near the antennas this may not be appropriate. Are there empty leads?

Smaller errors and confusing presentation

Does the paper really need extended data figures AND supplementary figures? Does that really improve the experience of the reader? This reviewer doubts it.

L17 In accord with .. This sentence does not describe an original finding of the paper, as this was already found in the 1990's

L19 The exploration ... techniques. This sentence is generic and does not belong in an abstract.

L34 "probability exceeds unity" Ouch!

L127 and on, the product of u and v gives t, but is this not a circular =reasoning, i.e.; has eq. 1 not been used implicitly in the determination of t? So it is not clear to me that the agreement actually says anything about the quality of the data,

Fig 2 is rather small.

The extended data figures are not cited in order and discussed too briefly in the text for them to be truly useful.

In conclusion, the manuscript deals with interesting matters but the title seems beside the point, the organization is a bit confusing and several questions remain on the validity of the data, especially in the light of the remarkable claims of detecting features orders of magnitude below the experimental noise, and the conclusions based thereupon. The manuscript text needs to be thoroughly revised before any decision can be made.

Version 1:

Reviewer comments:

Reviewer #1

(Remarks to the Author)

The authors have answered convincingly all comments/question and revised the manuscript accordingly. The revised version has been extensively modified and is improved. The additional figures and the ones that have been moved from the Extended Data to the main text offer better clarity to the main text. I recommend publication to Nature Communications in the current (revised) form

Reviewer #2

(Remarks to the Author)

The authors have provided thorough responses to all of our comments and have also addressed the concerns of reviewers 1 and 3. In particular, the addition of figures 3 and 4 greatly clarifies the role of noise in this experiment, illustrating how eigenvalue fluctuations may differ from the overall noise level allowing for the detection of closed channels via the TM. In conclusion, we recommend this manuscript in its present form for publication in Nature Communications.

Hui Cao, Rohin McIntosh

Reviewer #3

(Remarks to the Author)

Reviewer #4

(Remarks to the Author)

In this revision the authors have addressed many issues quite thoroughly and the result is a paper that is much clearer about their findings and the significance thereof. The discussion of the noise in the spectra which is critical to the question whether the TZs could be artefacts is much improved and I am happy to recommend this work for publication.

About the noise and drifts some specifics could still be added:

- I did not find the model (or noise characteristics) of the VNA in the paper.
- The origin of the drifts is left a bit open. The authors must have investigated what the most critical components are. For instance, did they exclude any effect of flexing the HF cables?

Some smaller remarks:

L:44 "Absence of a magnetic field" is confusing. In the absence of a magnetic field, Maxwell's equations do not allow for microwave propagation.

L73 seems a repeated mention of the particle diffusion model?

L74 latterly should probably be laterally.

95-98 This statement is, I think, dependent on the characteristics of the noise as indicated above, and cannot be stated as generally as here.

I advise the authors to consider the above remarks before publication.

RESPONSE TO REVIEWER'S COMMENTS

We thank the reviewers for their close reading of the manuscript and for their helpful and valuable comments, questions, and suggestions. These are all reflected in the revised manuscript. This has required an extensive modification of the text and the addition of three new figures in the main text.

Before responding to the specific points raised by the reviewers, we discuss the main changes made in consideration of the points raised by the reviewers:

- (1) The question was raised regarding the relevance of this work to conductivity and Ohm's law. The introduction now places greater emphasis on explaining the equivalence between quantum electronic and classical wave transport. In both realms, fluctuations, correlation, and Anderson localization play key roles. And, crucially, as Landauer argued, the dimensionless conductance is equivalent to the transmittance. However, the variation of conductance has not been studied as a function of sample width, particle energy or wave frequency, in a random diffusive ensemble in the crossover to a new channel. Here the classical equivalent is explored and the impact of all transmission eigenvalues has been explored.*
- (2) The ability to determine the properties of transmission eigenchannels even though their values are well below the noise level in the experiment is now explained. The noise level in different transmission eigenchannels is analysed. We show that correlation between transmission eigenchannels is part of the reason the sensitivity to noise of the lowest transmission eigenchannel is low enough that zeros can be observed. This is demonstrated with two new figures in the main text (Figs. 3 and 4).*
- (3) The proportionality of the modulation of the DOS and the conductance and the independence of the proportionality constant upon sample width points to global correlation. However, this finding was not mentioned by the reviewers. We believe the proportionality mentioned above is surprising in itself and another key sign of global correlation between the conductance modulation and the zeros and poles of the transmission matrix. We have therefore moved the figure demonstrating this point from the Extended Data figures to the main text (now Fig. 7).*
- (4) We now more emphatically present the evidence that we have indeed observed TZs, or, more formally, determined the properties of TZs from measurements of the TM:
(a) the demonstration that the minima in transmission of the lowest transmission eigenvalue has a clear quadratic spectrum about the minimum of single TZs, unobstructed by noise,
(b) the differentiation between single and conjugate pairs of TZs in the variation of the spectrum of transmission time of the lowest TE, t_N , with gain added to the raw experimental data,
(c) the presence of dips in t_N when absorption is present at precisely the frequencies of dips in τ_N , and,
(d) the correspondence of the spectra of τ_N and t_N with simulated spectra in which the zeros are shown in the complex frequency plane.*

The article now has 7 instead of 4 figures in the main text:

Figure 3 is a new figure. It shows the spectrum of the lowest transmission eigenchannel determined from the measured TM. The quadratic fit to $\tau_{62}(\omega)$ around the minimum is in accord with simulations. This shows that, even near the zero, the signal is not dominated by noise.

Figure 4 is also a new figure. It shows the impact of different levels of noise on the spectrum of the lowest transmission eigenvalue in simulations. The noise in the spectrum of the lowest transmission eigenchannel is seen to be much lower than the added noise for S/N ratios of 10^3 and 10^5 in the raw data. The variation of S/N with eigenchannel index n and with the level of noise is shown. This analysis shows that the correlation between TEs is the reason the noise is lower than the measurement noise. In addition, correlation between zeros and poles allow τ_N near a TZ to be observed at levels below the average noise level of the lowest transmission eigenvalue. Thus, one can find spectra of τ_N well below the experimental noise level.

Figure 7 has been moved from the Extended Data Figures to the main text. It shows the discovery of the correlation between dips in the conductance and peaks in the DOS. The identical proportionality constant at different widths in samples of the same length shows this is a real effect and that is not related to weak localization. This figure suggests that the density of poles determines the dips in conductance. It was already shown that the dip in conductance is related to the peak in the density of zeros. Thus, the poles and zeros are correlated. This figure points to a deep correlation within the TM and raises new fundamental theoretical questions.

Below we respond to all the points raised by the referees. We believe the paper is now sounder and more accessible and complete and its significance is clear.

We hope that, with the changes made, the paper can be accepted for publication in *Nature Communications*.

Reviewer #1

The authors of the current manuscript analyze experimentally and computationally the behavior of transmission eigenvalues in the proximity of a close-to-open channel threshold. They find that the minimal transmission eigenvalue vanishes at these frequencies, leading to dips in the transmittance of a diffusive sample. The experimental data are nicely compared with the numerical simulations and the main message is nicely conveyed. The results might be interesting for a large (and broad) scientific community – ranging from imaging to wireless communications.

The analysis relies on the connection between transmission eigenvalues with longitudinal energy density at the sample output and the longitudinal eigenchannel velocity. The equation that describes this connection is Eq. (1). Maybe I am missing something, but is this relation dimensionally correct?

Equation (1), $\tau_n = u_n v_n$, is dimensionally correct. As noted, above, u_n is the longitudinal energy density at the sample output, so its dimensions are $[E/L]$, where $[E]$ gives the dimensions of energy. To emphasize this point, we now add “which is the energy density per unit length,” after $u_n(L)$ in the sentence before Eq. (1). The dimensions on the left-hand and right-hand sides of Eq. (1) are the same: $[\tau_n] = E/s$ and $[u_n v_n] = [(E/L)(L/s)] = [E/s]$.

Also, I feel that a bit more work is needed in the introduction. While the abstract and conclusions were pretty clear, the introduction is missing this clarity. Specifically, the last paragraph has to

be re-written, the main result stated clearly and the potential connection to quantum electronics (and/or) other applications (e.g. imaging?) has to be highlighted.

We have now replaced the last paragraph of the introduction with two paragraphs. The first starts by pointing out that “Despite the intensive study of mesoscopic fluctuations in the conductance of single samples, measurements of the ensemble average of conductance in diffusive mesoscopic samples have not been reported.” It continues with the discussion of the findings of zeros in transmission in the present paper and their impact on conductance as a result of correlation of the TM.

In the second paragraph the connection to quantum electronics, to data compression, and to the applications of transmission and velocity zeros to ultrasensitive detection are discussed.

I am ready to recommend publication of the manuscript to Nature Communications provided that the authors address these comments in their revision.

We thank Reviewer #1 for highlighting precisely where the manuscript could be improved.

Reviewer #2

In this article, the authors experimentally measured low transmission eigenchannels through a scattering waveguide and demonstrated that modulations in the conductance g are driven by zeros of the transmission matrix (TM) near the crossover to a new channel. This study shows that for a small total number of channels, the zeros of the TM break Ohm's law, which asserts that conductance scales linearly with width. Although the existence of conductance fluctuations has been known for decades, the major achievement of this paper is the detection of low transmission channels from correlations in the transmission matrix and the experimental demonstration of the connection between these channels and conductance fluctuations.

Overall, the results are technically sound and the link to conductance variations makes this work both compelling and impactful. The experiments were done thoroughly, the data were collected over a large number of disorder realizations, and of high quality as result of measurement stability and a solid methodology. Aside from a few minor comments, I recommend this manuscript for publication in Nature Communications.

Comments and Suggestions:

The authors should make it clear that transmission eigenchannels below the noise level are never experimentally measured, but rather they are detected indirectly via measurements of the transmission matrix. Because of this, lines 58-59 and 267-268 come off as a bit misleading.

We thank the reviewer for pointing out that readers may misunderstand the use of the term “experimental measurement” here. To avoid any misunderstanding, we now explicitly state we have measured the TM and it is from this measurement that all properties of transmission are determined.

In this paper, we obtained the TM both from measurements and from simulations. In both cases, all properties of transmission are obtained by an analysis of the TM via the singular value decomposition in the basis of the modes of the empty waveguide. It is therefore natural for us to

say, when we are discussing result of the experiment, that we measured the transmission eigenchannels. It is worth noting that the use of the word “measured” is a question of familiarity since nothing is ever really measured. For example, it is natural for us to say that we measure the electric field in this experiment, but actually it is the output of the network analyser that we record. Similarly, it is natural for one to say that one measures the conductance, but actually one finds the ratio of the voltage and the current. From this we infer the conductance. Since the word “measure” might not be taken in the sense that was intended, we will use language that is less direct while stating that the TM was measured. Specifically, the word “determine” now appears frequently in the text.

In the present manuscript, we discuss the impact of noise in greater detail and therefore have eliminated the sentence in L267-268.

We have added Fig. 3 which shows the values of τ_N near a TZ. τ_N rises quadratically near an ultra-low minimum as expected for a TZ and does not vary randomly. The transmission of a single TZ determined from an analysis of the measured TM is not necessarily zero since it is displaced into the lower half of the complex plane by absorption. This can be compensated for by effectively removing the impact of absorption by adding gain as described in the supplementary notes. When the absorption in the medium is not perfectly uniform, the added gain necessary to effectively undo absorption is slightly different. But the process is limited because adding gain increases noise and so in some cases a TZ cannot be brought to the real axis while maintaining good quality spectra, as detailed in the discussion of Fig. 3.

We have also added Fig. 4 which shows the impact of added noise on τ_N in simulations. The average value of τ_N is observed at levels much smaller than the added noise. The variation of noise with eigenchannel index n and its impact on the S/N of the lowest TE is shown. Surprisingly, the standard deviation of the noise for different TEs vs. the eigenchannel index, n , falls on a Gaussian curve.

It would be helpful to know what the noise level is for the experiment. For example, what is the smallest transmission eigenchannel that can be experimentally measured if it is sent into the disordered waveguide?

The smallest value of transmission that appears in our determination of the spectrum of the lowest transmission eigenvalue seems to depend only upon the closeness of the measurement frequency to the frequency of the TZ and upon the absorption in the medium, which displaces the TZ from the real axis in the complex plane. A value of τ_N as low as 10^{-12} is shown in Supplementary Figure S1. We expect that the inferred value of τ_N could be arbitrarily lower if the measurement frequency were closer to the TZ.

For experimental results averaged over many disorder realizations (e.g., Fig. 1c) the authors should include the standard error in the plots. This will also provide some insight into the experimental noise.

The deviation of the signal level in Fig. 1c is not due to experimental noise but from mesoscopic fluctuations in the conductance in a single sample and the finite number of configurations averaged. We thank the reviewer for this suggestion and have added error bars in Fig. 1c. It can now be seen that the modulation we observe of the conductance is larger than the noise.

We note that the signal in Fig. 1c for $L = 23$ cm is smoother (less noisy) than that for the other lengths. This is primarily because measurements were made for a larger number of configurations for this sample. For diffusive samples, $\text{var}(g)$ in a single spectrum is a constant, $\text{var}(g) \sim 2/15$, giving the standard deviation of $\sigma = \sqrt{2/15} = 0.37$ due to universal conductance fluctuations. The standard deviation in the average the conductance over M different configurations is, $\sigma_M = \sqrt{\frac{2}{15}}/\sqrt{M} = 0.37\sqrt{M}$. The error bars now introduced in Fig. 2c bracket the curves shown of g with $\pm\sigma/\sqrt{M}$. $M = 23, 6, 6$ for ensembles with lengths, $L = 23, 40, 61$ cm.

Conceptualizing zeros and poles of the TM can be challenging, as low transmission eigenchannels are typically not realizable in experiments. Including a physical description for how these channels behave would be helpful, even if this behaviour is not explicitly observable.

It was the experimental measurement of the TM that yielded the properties of the low transmission eigenchannels on the output and input surfaces. For example, the zeros of the EV were discovered in our measurements, as was the possibility of determining the spectrum of the lowest TE for a waveguide with up to 64 channels.

The effect of poles on transport is particularly confusing. Can poles be brought to the real axis with sufficient gain and, if so, what would it physically mean for the transmission to diverge? Is this related to lasing as in the case of a pole for the scattering matrix?

Yes, the transmission would diverge when a pole is brought to the real axis. The poles of the TM are the same as the poles of the scattering matrix.

The zeros are different. Whereas the zeros of the scattering matrix are conjugate to the poles TZs either lie on the real axis or appear as conjugate pairs when there is no dissipation. In addition, the TZs are topological in the sense that single zeros remain on the real axis and conjugate pairs of TZs remain so as a sample is deformed. However, two single zeros and a conjugate pair can interconvert.

The current title is somewhat confusing to general readers. Specifically, it is unclear what the zeros and poles refer to, making it hard to determine what this paper is about or worth reading. The authors are encouraged to provide a clearer title than the current one.

*In accord with the referee's suggestion, we have changed the title to **Ohm's law lost and regained: observation and impact of transmission and velocity zeros**. It is now clear that we treat two types of zeros, transmission zeros and eigenchannel velocity zeros.*

I would like to note that the beginning of the title is important because this paper explains the departures from Ohm's law in small samples and the approach to Ohm's law in larger sample. This is a major issue in mesoscopic physics whose full explanation has not been given previously. The conductance through a crossover to a new channel in a diffusive medium had not been measured previously. This work shows how it is crucial to include the wave nature of transport to explain the scaling of conductance. This represents a new mesoscopic effect that we believe will be crucial in understanding the nature of electronic conductance and classical wave

propagation. The phrase “lost and regained” in the title is a reference to John Milton’s two epic poems Paradise Lost and Paradise Regained.

Reviewer #3 (Remarks to the Author):

We thank Reviewer’s #2 and 3 for their careful consideration of our paper and for their valuable suggestions.

Reviewer #4 (Remarks to the Author):

The work by Joshi et al is a very interesting exploration of the occurrence, in microwave experiments as well as in numerical simulations, of near-zero eigenvalues of the transmission matrix in the diffusive regime.

Typically, the smallest eigenvalue of the TM is expected to be exponentially small and negligible with respect to the dominating eigenchannels which have transmittance of order unity. Here the authors show that near the onset of a new channel dips in the transmittance of this lowest channel occur to yield an even lower transmittance. The authors were not able to directly excite such channels to demonstrate the absence of transmitted power directly, the low transmittances were the result of an eigenvalue decomposition of the measured transmission matrix.

Here we find an unexplained and unelaborated weakness of the paper: The authors claim to resolve a channel with a transmittance that is nine orders of magnitude below the experimental signal to noise. They give no clue how this is possible and why we should not immediately discard the effect as an artefact of their data analysis procedure.

In the revised manuscript, we give additional critical support for the reliability of the spectra of τ_N determined from measurements of the field transmission matrix even though the values of transmission near TZs fall far below the noise level in the measurement. To analyse the noise in a determination of the TM and to demonstrate that we have indeed observed a TZ, we have added Fig. 3 which fits a quadratic form found in numerical simulations to the spectrum of τ_N to the spectrum of τ_N around the minimum value of τ_N . We have also added Fig. 4, in which noise is added to the simulations, to show that noise in the spectrum of τ_N is much smaller than the noise added to the TM in simulations. The identification of dips in τ_N with TZs, and so the accuracy of the measurement of the lowest TE is further confirmed by the dramatic variation of the transmission time of the lowest TE, t_N , with the addition of different amounts of gain to the experimental data. In addition, the vanishing of transmission at channel crossovers is observed and explained as due to velocity zeros, which do not self-average.

Moreover, their determination of the experimental SNR leads to many other questions. They claim an average value of 1300, but the 3 typical curves they show all have a SNR of 200 or less. And what is the average SNR anyway, $\langle N \rangle$, $\langle S/N \rangle$, or $1/\langle N/S \rangle$?

It seems that the procedure they use to determine the SNR mostly measures drifts due to e.g. mechanical or thermal creep in the sample, not the instantaneous SNR of the measurement device. So very likely this is a highly correlated form of noise, that could well cause artefacts in the data. A result of such artefacts could well be that transmission zeros that were originally not on the real axis are shifted onto it.

We now give a more extensive and precise discussion of the noise. We discuss that, in one case we measured parts of the TM of one of the 23 sample configurations a second time. Averaging over at the repeated spectra give a S/N ratio of 280. The procedure used to estimate the S/N is illustrated in Supplementary Figure S4.

While the experimental work is very interesting, there more are several statements in the manuscript that seem wrong, contradictory, circular, or imprecise, and that make me reluctant to advise in favor of publication. Therefore, I recommend a thorough revision of the text.

Comments and questions on the procedure

First of all it should be explained how a signal that is nine (or six?) orders below the noise can be trusted. What checks did the authors do to make sure that transmission zeros really exist and are not artefacts of the noise/drifts that occur during their measurements?

We have mentioned above and now emphasized in the manuscript the checks we have performed that show that TZs really exist. We show (1) the spectrum of τ_N around the minimum in transmission (Fig. 3), in which the quadratic form is not obscured by noise, (2) the insensitivity of spectra of τ_N to added noise in the simulations (Fig. 4), (3) the presence of dips in t_N at precisely the frequencies of dips in τ_N , (4) the variation of t_N with added absorption that behave in accord with theoretical predictions and simulations, and (5) the sharp drop in transmission in the two new TE that appear precisely at the channel crossover where the EV of the new channels is found to vanish. All these checks indicate that we are observing a real effect and not an artefact.

Thus even in the presence of drift in the data, we can conclude that at the levels present in our experiment zeros of transmission to TZs and EVs can be clearly identified and their impact upon transport can be reliably studied.

What is the claim on the noise level anyway? L 267 speaks of 9 orders below the experimental noise level, while L 99 it is 9 orders between the signal level of the most open channel, and the SNR is claimed to be about 1000. So there is a factor 1000 inconsistency between these statements.

In Fig. 2b, the lowest value of τ_N shown in a sample of length 23 cm is 10^{-9} , whereas in Supplementary Fig. S1 in a sample of length 40 cm, τ_N drops to 10^{-12} . The lowest value that can be achieved may be demonstration limited by the frequency step in the experiment of 300 kHz. This is now clarified in the text.

In the case of the simulations, the reference for the recursive Green's function method (42) does

not seem to be the original one (45). This method, although powerful, can have numerical “gain” or “loss” due to rounding errors, did the authors check the unitarity of the S-matrix?

Reference 42 appeared before the earlier primary reference 45 in the original submission because reference 42 had appeared earlier in the text. We have now removed reference 42 at this point. We have now checked the unitarity of the scattering matrix

L102 “measurements of dips in transmission well below the experimental noise level reflect correlation between the eigenvalues” That sounds nonsensical as only one eigenvalue seems affected. They DO however reflect correlations between the matrix elements that seem to (through some unexplained mechanism?) survive the noise.

We have shown in what is now Fig. 6c and in Extended Data Fig. 6 that, when the transmission in the lowest eigenchannel drops towards zero near a transmission zero, all the transmission eigenvalues are correlated. It is as a result of this correlation that a dip appears in the conductance.

The insensitivity to noise is the result of correlation of the transmission eigenvalues.

The discussion on Ohms law (L255 and on) seems to apply just as well to the type of conductance modulation seen in fig 4a and already described long ago (e.g.

DOI:<https://doi.org/10.1103/PhysRevB.40.5941>).

The modulation of the transmittance we have measured was found previously in simulations, but the role of TEs and particularly the lowest-transmission TE is not mentioned in the article by Chu and Sorbello mentioned above nor in any other publication. The dips in this article are attributed to “multiple scattering between the impurity and the waveguide walls.”

In this work, we have (1) made the first measurement of the transmission of the lowest TE that allows the observation of zeros of transmission, (2) made the first measurement of both single and conjugate pairs of TZs and of velocity zeros, (3) discovered the correlation between transmission eigenvalues that allow the lowest TE to affect the conductance, and (4) demonstrated the correlation between zeros and poles of the TM.

We will include the important reference given above and provided a more extended discussion of simulation that show drops in conductance. In past work, this was not related to zeros of transmission. In addition, there has been no measurements of drops in electronic conductance in diffusive media in which the dimensionless conductance is larger than unity.

Ohm’s law has nothing to say about single transmission channels.

The conductance is the sum of the transmission eigenvalues, so it is rather the eigenchannels that have something to say about Ohm’s law. Transmission zeros were only predicted three years ago and their impact on the conductance had not been anticipated in previous literature. Zeros of EVs not been discussed previously in a multichannel random system had. This paper shows the impact arising from correlation between the transmission eigenvalues is real and the source of the modulation of conductance.

Hence this discussion (and the first part of the title of the paper) seem beside the point. It is however interesting that the conductance drops occur around the same time as the transmission zeros.

*We believe the first part of the title with its reference to the violation and the restoration of Ohm's law is essential because this work opens up a new chapter in the study of the mesoscopic conductance. The field of mesoscopic physics arose around observations of fluctuations in conductance, which are ripples on the conductance. But the impact of coherent waves on the **average** conductance and its scaling around a crossover to a new channel was not observed experimental. We have done this here.*

L67 .. the continuous field can be recovered ... This suggests full recovery of the field is possible but L320 "the TM is not complete".

The discussions on lines 67 and 320 of the original manuscript are of two different matters. In the earlier discussion, the spatial profile of the electromagnetic field can be determined via the Whittaker-Shannon 2D sampling theory from the measurement of the field at a finite number of points. In the later discussion, it is the recovery of the transmission matrix that is the issue. We find here, that zeros of transmission occur even for an incomplete TM. Below a certain degree of completeness, transmission does not vanish. We also note that because of incompleteness of the TM we do not observe perfect transmission.

How incomplete is the TM and how does this incompleteness relate to what is claimed on L67?

One measure of the incompleteness is the highest transmission coefficient that can be achieved in the highest transmission eigenvalue. For the sample with $L = 23$ cm, the highest transmission, once the impact of absorption is effectively removed is 0.8.

In the same line: The authors express the field in terms of the propagating modes of the empty waveguide. However, if scatterers are near the antennas this may not be appropriate. Are there empty leads?

Yes, a 2-cm length of empty waveguide was appended on the front and back sides of the sample to avoid the problem pointed out by the reviewer. This should have been mentioned, but was not. It is now discussed in the experimental section of Methods.

Smaller errors and confusing presentation

Does the paper really need extended data figures AND supplementary figures? Does that really improve the experience of the reader? This reviewer doubts it.

The figures in all parts of the paper are used in clarifying the questions raised by the reviewers, and are therefore important. We now more explicitly make clear the connection of each figure to the thrust of the paper.

L17 In accord with .. This sentence does not describe an original finding of the paper, as this was already found in the 1990's

We had referenced previous simulations showing dips in conductance. In the revised manuscript, we have written a separate paragraph on earlier findings, In our paper we show precisely how the classical limit is approached as a result of the widening of the distribution of TZs and the finite range in TE index of correlation of transmission eigenvalues.

L19 The exploration ... techniques. This sentence is generic and does not belong in an abstract.

We have now taken out this sentence. We believe the importance of the paper is expressed in the abstract without this sentence.

There is a generic feel to this sentence, but we feel that this paper really does open a new door to the study of mesoscopic physics. We have for the first time measured all the eigenvalues of the TM and thereby measured and explained the nature of the scaling of conductance with sample width in terms of a totally new set of ideas relating to the transmission and velocity zeros, and to the deep correlation within the TM, which among other things leads to a totally unexpected quantitative correlation between peaks in conductance and dips in the density of states. The variety of zeros of transmission observed here for the first time open up new possibilities for ultrasensitive detection.

L34 "probability exceeds unity" Ouch!

Yes! Thanks for pointing this out. We have replaced the sentence with "Once the number of such crossings of a typical coherence length exceeds unity, waves are exponentially localized within the sample and conductance falls exponentially"¹⁵⁻²⁰.

L127 and on, the product of u and v gives t, but is this not a circular =reasoning, i.e.; has eq. 1 not been used implicitly in the determination of t? So, it is not clear to me that the agreement actually says anything about the quality of the data,

We thank the referee for raising this point. A calculation of this point shows the referee is correct! The calculation is given below and is included in the Supplementary Notes.

The basis function corresponding to the m^{th} mode of the empty waveguide is φ_m . Then the field coefficient of transmission for the field in position space is the Greens function G . Including all the polarizations, G is projected into the space of waveguide modes according to $t = 2i\varphi e \varphi^\dagger$, where t is now the transmission matrix in the basis of waveguide modes and φ is the vector of m waveguide modes.

The singular value decomposition of transmission matrix in the space of waveguide modes is: $t = UAV^\dagger$, where V and U are singular matrices corresponding to the input and output surfaces, respectively. The transmission eigenvalues τ_n are the squares of the singular values, which are the diagonal elements of Λ : $\tau_n = |\lambda_{nn}|^2$. The eigenchannel velocities v_n are determined by $\frac{1}{v_n} = \sum_m \frac{1}{v_m} |u_{nm}|^2$, where u_{nm} is an element of U and v_m is the group velocity of the m^{th} waveguide

mode. The eigenchannel energy densities u_n are given by $u_n = \sum_m \frac{1}{v_m} |t_{nm} v_{mn}|^2$, where t_{nm} is an element of t and v_{mn} is an element of V . Since $tV = U\Lambda$, this implies that $u_n = \sum_m \frac{1}{v_m} |\lambda_{nm} u_{mn}|^2 = \tau_n \sum_m \frac{1}{v_m} |u_{nm}|^2 = \frac{\tau_n}{v_n}$.

Fig 2 is rather small.

We have now enlarged the figure and made it easier to read by running the frames for a single configuration and for the average over configurations vertically. This way the frequency variation of all quantities can be seen at a glance and only a label for the frequency axis is needed for each set of the four frames.

The extended data figures are not cited in order and discussed too briefly in the text for them to be truly useful.

The figures are now numbered correctly and the importance of all figures is made clear in the text.

In conclusion, the manuscript deals with interesting matters but the title seems beside the point, the organization is a bit confusing and several questions remain on the validity of the data, especially in the light of the remarkable claims of detecting features orders of magnitude below the experimental noise, and the conclusions based thereupon. The manuscript text needs to be thoroughly revised before any decision can be made.

We thank Reviewer #4 for pointing out apparent inconsistencies and places where greater clarity was required. We believe we have provided this clarity in the revised manuscript in accordance with the remarks of all the reviewers. We hope that the changes made in response to the points raised by the referees will make this article acceptable for publication in Nature Communications.

We note that none of the reviewers noted the figure showing in detail the proportionality of the modulation of the conductance to that of the DOS. Unlike the qualitative discussion of the correlation of zeros of transmission with a depression in the conductance, the connection here is quasi-quantitative. "Quasi," because there is some subjectivity in judging departures from the regular secular increase of the conductance and DOS with increasing sample width. However, the constant of proportionality is independent of width and it decays with length. This effect shows a correlation of the DOS (poles) and the conductance and thus between the density of zeros and poles. This represents a correlation between the zeros and poles, which is embedded in the TM and a key element of the global correlation within the TM. Presumably, the knowledge of the position of the poles in complex frequency space is embedded in the TM. Since the position of the poles determines the position of the zeros. Working this out quantitatively is a major challenge that may help unlock the store of relationships encapsulated within the TM. We have therefore moved Extended Data Figure 4 to the main text as Fig. 7.

REVIEWERS' COMMENTS

Reviewer #1 (Remarks to the Author):

The authors have answered convincingly all comments/question and revised the manuscript accordingly. The revised version has been extensively modified and is improved. The additional figures and the ones that have been moved from the Extended Data to the main text offer better clarity to the main text. I recommend publication to Nature Communications in the current (revised) form

Reviewer #2 (Remarks to the Author):

The authors have provided thorough responses to all of our comments and have also addressed the concerns of reviewers 1 and 3. In particular, the addition of figures 3 and 4 greatly clarifies the role of noise in this experiment, illustrating how eigenvalue fluctuations may differ from the overall noise level allowing for the detection of closed channels via the TM. In conclusion, we recommend this manuscript in its present form for publication in Nature Communications.

Reviewer #3 (Remarks to the Author):

Reviewer #4 (Remarks to the Author):

In this revision the authors have addressed many issues quite thoroughly and the result is a paper that is much clearer about their findings and the significance thereof. The discussion of the noise in the spectra which is critical to the question whether the TZs could be artefacts is much improved and I am happy to recommend this work for publication.

About the noise and drifts some specifics could still be added:

- I did not find the model (or noise characteristics) of the VNA in the paper.
- The origin of the drifts is left a bit open. The authors must have investigated what the most critical components are. For instance, did they exclude any effect of flexing the HF cables?

We now give a fuller discussion of the sources of noise in our experiment in the second and third paragraphs of Supplementary note 4, which follows:

The principal factors that determine the SNR in our experiments are the variation with temperature of the gain of the amplifier and of the structure of the sample, and the accuracy of the positioning of the source and detection antennas. Since the transmitted field is a result of interference of partial waves following trajectories of different lengths, changes in path lengths of transmitted waves with temperature lead to changes in the transmitted intensity. The interference of waves is affected by increasing temperature via the expansion of the elements of the sample of alumina spheres in Styrofoam shells in air contained in a copper tube. This is partially compensated for by the associated reduction in the indices of refraction of media of lower density as temperature increases. In addition, the changes in the sample lead to changes in scattering, and so in the distribution of wave trajectories.

The typical pathlength of transmitted waves may be found from the average transmission time through the 23-cm-long sample given by the average of the phase derivative, $\langle t \rangle = \langle d\phi/d\omega \rangle = 34$ ns. Since sharp resonances do not occur over the spectral range of the measurements, the average refractive index may be taken as the average of the indices of refraction weighted by their volume fraction, to give an average index of $n = 1.15$. This gives an optical path length of $\langle c/n \rangle \langle t \rangle = 8.9$ m. These factors result in a change in the partial waves and the interference between them, which cannot be calculated readily, but likely contributes to the variation of spectra over time. Another possible source of noise in the measurement of the TM is shifting of the sample, but this seems to have been eliminated by shaking the sample before the measurements are made. Finally, the control of the position of the source and detector antennas are not perfect. Most of these factors can be improved with better control of the temperature and

positioning. Nonetheless, it was possible to reliably obtain spectra of transmission eigenvalues because of the correlation between the transmission eigenvalues.

Some smaller remarks:

L:44 "Absence of a magnetic field" is confusing. In the absence of a magnetic field, Maxwell's equations do not allow for microwave propagation.

We have now made clear this is the applied magnetic field in a measurement of the electronic conductance, not the electromagnetic field by adding the word "applied."

L73 seems a repeated mention of the particle diffusion model?

The first mention of particle diffusion was to explain that this leads to Ohm's law. The second mention related to the experimental demonstration of an equivalent of Ohm's law for photons, as opposed to electrons.

L74 latterly should probably be laterally.

Thanks. Yes "laterally"

95-98 This statement is, I think, dependent on the characteristics of the noise as indicated above, and cannot be stated as generally as here.

We are saying here that it is possible to measure the TM at values of transmission eigenvalues that are well below the noise level in the measurements. This is just what is done here. It is certainly possible that there are cases when it will be impossible to obtain spectra of all the transmission eigenvalues, but that is not the case considered here.

I advise the authors to consider the above remarks before publication.